# Grounding Long-Context Reasoning with Contextual Normalization for Retrieval-Augmented Generation

## Abstract

Retrieval-Augmented Generation (RAG) has become an essential approach for extending the reasoning and knowledge capacity of large language models (LLMs). While prior research has primarily focused on retrieval quality and prompting strategies, the influence of how the retrieved documents are framed, i.e. context format, remains underexplored. We show that seemingly superficial choices, such as delimiters or structural markers in key–value extraction, can induce substantial shifts in accuracy and stability, even when semantic content is identical. To systematically investigate this effect, we design controlled experiments that vary context density, delimiter styles, and positional placement, revealing the underlying factors that govern performance differences. Building on these insights, we introduce Contextual Normalization, a lightweight strategy that adaptively standardizes context representations before generation. Extensive experiments on both controlled and real-world RAG benchmarks across diverse settings demonstrate that the proposed strategy consistently improves robustness to order variation and strengthens long-context utilization. These findings underscore that reliable RAG depends not only on retrieving the right content, but also on how that content is presented, offering both new empirical evidence and a practical technique for better long-context reasoning.

## 1 Introduction

Retrieval-Augmented Generation (RAG) has emerged as a foundational paradigm for enabling large language models (LLMs) to scale to knowledge-intensive tasks by conditioning generation on external documents retrieved from large corpora (Lewis et al., 2020b; Borgeaud et al., 2022). In a standard pipeline, a retriever first identifies potentially relevant texts for a given query, and these documents are then concatenated into a prompt for the LLM. With the advent of long-context LLMs that can process tens of thousands of tokens (Xiao et al., 2024; Xu et al., 2024), the opportunities for complex reasoning over vast information spaces have been unlocked, making RAG increasingly central to real-world applications such as open-domain QA and scientific literature analysis.

While long-context extensions enable RAG systems to scale to much larger evidence pools, they also introduce new challenges. Recent study (Leng et al., 2024) highlights these limitations by systematically varying context length, from 2K up to 128K tokens across dozens of models, and documenting consistent failure modes when contexts become too long or unwieldy. With the number of retrieved chunks increasing, LLMs face amplified retrieval noise, redundancy across overlapping documents, and dilution of truly relevant evidence. These issues often make it harder for LLMs to distinguish signal from distraction, leading to unstable reasoning and degraded accuracy. Moreover, positional biases (Liu et al., 2024a; Zhang et al., 2024) further interact with these challenges: LLMs tend to over-attend to the beginning or end of a prompt, leaving evidence buried in the middle underutilized. Together, these factors expose a fundamental brittleness in long-context RAG that limits its reliability in real-world deployments.

To mitigate these limitations, a growing line of research has explored strategies to improve long-context RAG performance. One representative approach is the prompt optimization (Liu et al., 2024b), where multiple permutations of retrieved chunks are scored, and the prompt yielding the

highest likelihood is selected for answering. Another direction relies on synthetic supervision: An et al. (2024) propose constructing curated datasets where answers depend on specific chunks within extended inputs, encouraging LLMs to develop position-invariant reasoning strategies. While effective in controlled settings, such methods face scalability issues, as generating chunk-level annotations is costly and risks positional overfitting. Architectural modifications, such as redesigned positional encodings (Zhang et al., 2024), offer more fundamental solutions but require non-trivial changes to model internals. Complementary work (Vladika & Matthes, 2025) provides further evidence that context size, snippet count, and model architecture interact in subtle ways, jointly shaping robustness and accuracy.

To benchmark long-context reasoning in a way that isolates potential factors from prior knowledge, Liu et al. (2024a) propose the key–value extraction task, where LLMs must retrieve the correct value for a given key from a synthetic context. Inspired by this controlled setup, we extend the analysis and uncover a striking finding. As illustrated in Figure 1, even when semantics and input length are held constant, altering the surface format of key–value pairs, for instance, representing them as UUIDs, plain texts, or switching delimiters such as "-" versus "&", leads to substantial performance differences. This finding highlights that the presentation format of context, beyond its

size or order, plays a critical role in determining long-context reasoning performance. It also sheds light on a possible research direction: if the surface format of context can be altered while preserving semantics, could long-context RAG performance also be systematically improved? Therefore, we propose the Contextual Normalization (C-NORM), a lightweight and model-agnostic framework designed to enhance long-context RAG performance by adaptively reformatting the input context. Rather than introducing new supervision or modifying model architectures, C-NORM leverages the insight that the surface format of retrieved documents directly influences how LLMs allocate attention and ground their reasoning. By systematically evaluating candidate formatting strategies using the proposed Attention Balance Score, C-NORM automatically selects the representation that promotes balanced and semantically aligned attention. This design enables LLMs to reason more robustly across long inputs, without requiring architectural changes, retraining, or costly annotation.

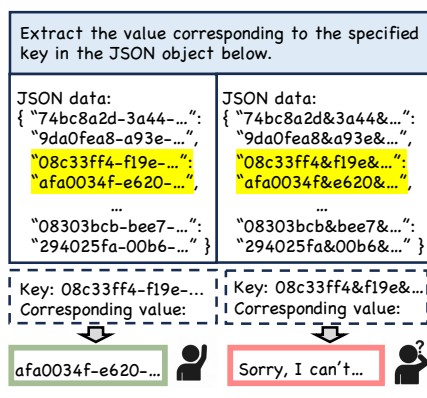

Figure 1: Illustration of different formats yield substantial differences in RAG performance.

**The contributions of this work** can be summarized as follows:

- We highlight the often-overlooked but critical role of context format in long-context RAG, demonstrating that even seemingly superficial representational choices can substantially alter both robustness and reasoning capacity. To explain this phenomenon, we propose two underlying factors, i.e. tokenization and attention allocation, that account for the sensitivity of LLMs to format variations. We validate these hypotheses through targeted experiments in controlled settings.

- We propose C-NORM, a principled approach that reformulates context presentation as a normalization problem. By leveraging attention attributions as a selection criterion, C-NORM adaptively chooses the most effective model-aware format, offering a simple, plug-and-play solution.

- We conduct extensive experiments under both controlled and real-world settings, demonstrating that C-NORM consistently improves the RAG performance across diverse models. Gains are especially pronounced in challenging long-context scenarios, underscoring its practical value for reliable long-context RAG.

## 2 How Context Format Grounds

The performance of RAG systems in long-context scenarios is profoundly influenced by the effective integration of retrieved information (Borgeaud et al., 2022; Karpukhin et al., 2020). While previous work (Asai et al., 2023) has primarily focused on the quantity and relevance of retrieved documents, we posit that the internal format of this information, more specifically, how context content in each

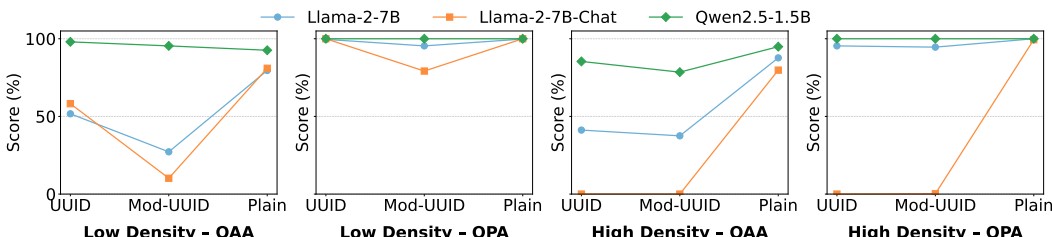

Figure 2: Model performance on key-value extraction task.

chunk is structured, plays a critical role in grounding the model's generation. To investigate this hypothesis, we design a set of experiments centered on the key-value extraction task (Liu et al., 2024a), a canonical challenge for LLMs requiring precise information retrieval from a given context. The goal of the task is to retrieve the value of a specific key from a long JSON object. More details are provided in Appendix A.

**Context Formats.** To systematically investigate how context format grounds LLM's generation, we propose the following context formats in the experiments: **UUID**, **Plain Text**, and **Modified UUID**. Each format varies only in its use of structured identifiers, allowing us to analyze the model performance to metadata and special characters. The Universally Unique Identifiers (UUIDs) is a 128-bit number used to uniquely identify information in computer systems, which is utilized in the standard key-value retrieval task (Liu et al., 2024a). They are typically represented as a 32-digit hexadecimal number, displayed in five groups separated by hyphens. In Plain Text, all structured identifiers are removed. Both key and value are flattened into a continuous text string. Modified UUID introduces a subtle change to the original UUID by replacing the hyphen (-) with a different delimiter (&). This simple substitution allows us to probe the sensitivity of LLMs to variations in structured data, revealing how the model processes the context.

**Settings.** We design a controlled experiment with 500 samples to evaluate the impact of context format on long-context RAG. We permute the position of a "gold" key within a long context, then measure the performance of three LLMs: LLaMA-2-7B (Touvron et al., 2023), LLaMA-2-7B-Chat (Touvron et al., 2023), and Qwen2.5-1.5B (Yang et al., 2024). The experiments are conducted with two context configurations to test different densities: low-density (40 contexts with 32 characters each) and high-density (10 contexts with 128 characters each). For each setting, we record two key metrics: the Overall Averaged Accuracy (**OAA**) across all positions, which measures robustness to all gold key positions, and the Optimal Positioned Accuracy (**OPA**), which captures the model's best-case performance under ideal position of gold key.

**Analysis.** As shown in Figure 2, the LLM performance is highly sensitive to the format of the retrieved context. The LLaMA models consistently achieve their best results with Plain Text. In contrast, Qwen2.5-1.5B excels with the UUID format in low-density settings, but shifts to favor Plain Text in high-density settings. This divergence across models indicates that no single format is universally optimal, as model behavior depends heavily on its internal dynamics. Extending the context window can improve performance, but the results make clear that it does not mitigate format effects. Even with Qwen's 128k-token window, format remains a decisive factor. Notably, Qwen's advantage with UUIDs holds only in low-density settings (32-character chunks). In high-density settings (128-character chunks), its UUID performance (0.854 OAA) is surpassed by Plain Text (0.949 OAA), mirroring the LLaMA family's preference. The results also illustrate how fragile models can be to minor format changes. Simply replacing a hyphen with an ampersand in the Modified UUID format can cause LLaMA-2-7B-Chat's OAA to collapse from 0.810 to 0.102. In some high-density structured cases, the model even refused to answer entirely. Overall, these findings underscore that the format of retrieved context is not a neutral choice but a critical factor that can either stabilize or destabilize LLM performance in RAG.

## 3 UNPACKING THE GROUNDING MECHANISM.

To understand why context formats affect long-context reasoning, we delve into the internal dynamics of different LLMs. We analyze two complementary perspectives, i.e. tokenization and the

distribution of attention, which govern how the model allocates focus across positions. Together, these analyses reveal how subtle choices in context shape robustness and reasoning capacity.

## 3.1 TOKENIZATION

We first look into the impact of tokenization on this key-value extraction task, focusing specifically on how delimiter choices interact with the tokenizer internals. For LLMs such as Qwen2.5, which use a SentencePiece-based tokenizer (Kudo & Richardson, 2018), delimiter characters such as '-', ':', '&', '_', and '+' affect the token count of the input string significantly. To this end, we use 200 synthetic key-value samples, each consisting of 40 context pairs. The target (gold) key-value pair is inserted at each position. We report OAA as the aggregated metric. As shown in Figure 3, the results reveal a relatively strong negative correlation (Pearson's $r = -0.82$) between the number of tokens produced and the corresponding OAA. In other words, delimiters that yield shorter tokenized sequences (e.g., hyphens or colons) lead to higher accuracy, while those producing longer tokenizations degrade performance. This suggests that more compact representations enable LLMs to allocate attention more effectively within the fixed context window. However, this behavior is not universal. For LLMs like LLaMA-2, which tokenize many symbols (e.g., -, _, /, +) into single-character tokens, the number of tokens remains unchanged across different delimiters. In these cases, performance still varies with different delimiters, but the effect cannot be attributed to token count.

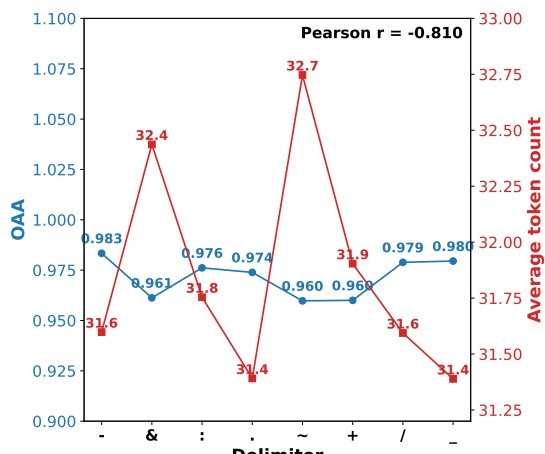

Figure 3: Qwen2.5-1.5B performance across delimiter configurations. Across settings, we observe a negative trend: configurations that inflate tokenization length tend to yield lower OAA.

## 3.2 ATTENTION ATTRIBUTION.

To further understand how context format shapes long-context reasoning, we use the low-density setting to observe last-layer attention distributions in both LLaMA-2-7B and Qwen2.5-1.5B, aiming to understand why different context formats lead to different performance patterns across models. Specifically, we construct 20 key–value pairs and place the target key at varying positions and measure how attention from the final token is allocated across the sequence under both UUID and Plain Text. Figure 4 presents the attention weights from the final token to all preceding tokens. For Qwen2.5-1.5B, the Plain Text format yields sharp attention peaks at the beginning and end of the sequence, while the UUID format produces a more uniform distribution, with increased emphasis on middle positions. On the contrary, in LLaMA-2-7B, UUID contexts concentrate attention at the sequence boundaries, whereas plain-text contexts lead to stronger coverage of the middle portion. This contrast in allocation explains the opposite performance trends observed in Table 2: formats that encourage more balanced attention across the sequence tend to achieve higher robustness and overall accuracy in long-context retrieval.

**On the Role of Training Data.** To further probe why different context formats lead to distinct attention allocation patterns, we attempt to trace the effect back to the training data. With Stanford-Alpaca-7B (Taori et al., 2023), we sort tokens in its fine-tuning corpus by frequency of occurrence, and then reconstruct QA contexts where original tokens are replaced with either the most frequent or least frequent tokens. This design tests whether exposure frequency in fine-tuning data influences how attention is distributed across contexts. However, the results do not show a clear relationship between token frequency and LLM performance or attention allocation. This indicates that the grounding mechanism behind context-format sensitivity is more complex than simple token frequency statistics, likely shaped by deeper patterns acquired during both pretraining and fine-tuning. We provide the details of the experiment in Appendix B.

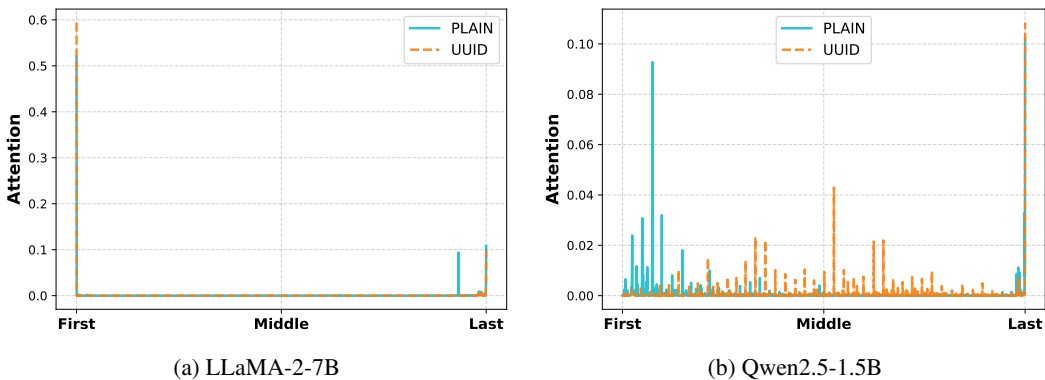

(a) LLaMA-2-7B                                    (b) Qwen2.5-1.5B

Figure 4: Attention attributions in long-context reasoning under low-density settings. The x-axis denotes the position of input tokens.

## 4 CONTEXTUAL NORMALIZATION FOR ENHANCED RETRIEVAL-AUGMENTED GENERATION

Inspired by the above findings, we propose Contextual Normalization, C-NORM, a lightweight procedure that standardizes retrieved passages into a format that better supports grounding in long contexts. As shown in Figure 5, the method operates in three stages: (i) candidate formatting of contexts, (ii) attention-guided scoring to select a format, and (iii) application of the chosen format for all contexts in RAG. This procedure is model-aware yet training-free, requiring only a forward pass with attention outputs.

### 4.1 CANDIDATE FORMATTING

Given a query $q \in Q$ and a set of retrieved passages $\mathcal{D} = \{d_1, \ldots, d_m\}$, we generate format variants of each passage using sentence-level restructuring. Specifically, with a delimiter $f \in \{\texttt{none}, \texttt{-}, \texttt{\_}, \texttt{:}, \texttt{.}, \sim, \texttt{+}, \texttt{/}, \texttt{\&}, \ldots\}$ and the predefined ratio $p \in [0, 1]$, a fraction $p$ of sentences in $d_i$ are reformatted by replacing whitespace with $f$. This procedure preserves semantic content while varying structural cues in a controlled manner, creating candidate contexts $\tilde{d}_i^{(f,p)}$. The formatted documents are then assembled as contexts into prompts for finishing the task.

### 4.2 ATTENTION-GUIDED SCORING

To assess which format best supports grounding, we propose an **Attention Balance Score (ABS)** from the LLM's internal attention distributions. For each candidate format $f$, we sample a subset of prompts $S$ with $|S| \ll |Q|$, and extract the last-layer attention vector $a \in \mathbb{R}^T$ corresponding to the final token. We then compute:

$$\text{ABS}(a) = 1 - 2 \cdot |\mu - 0.5|, \quad \text{where } \mu = \sum_{t=1}^{T} (\frac{t-1}{T-1}) \cdot \frac{a_t}{\sum_j a_j}.$$

This score peaks when attention mass is balanced across the sequence, avoiding pathological focus on only the beginning or end of the input. The final delimiter $f^\star$ is chosen by maximizing the average ABS across $S$ sampled prompts:

$$f^\star = \arg\max_f \frac{1}{S} \sum_{s=1}^{S} \text{ABS}(a_s^{(f)}).$$

### 4.3 FORMAT APPLICATION

At inference time, all sentences in the retrieved documents are reformatted with the selected configuration $(f^\star, p)$ before constructing the final prompt. Here, $f^\star$ denotes the delimiter format that has

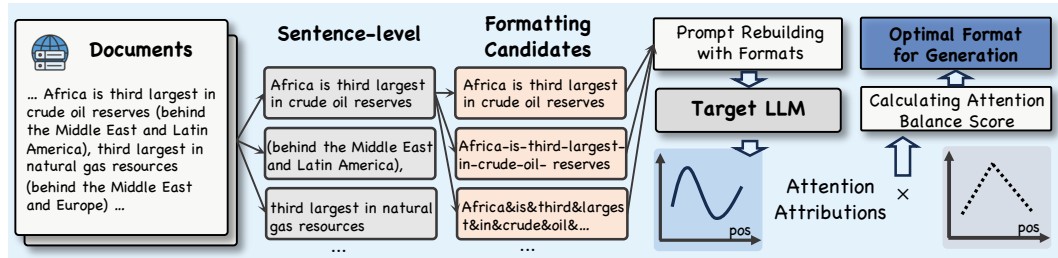

Figure 5: Overview of the proposed C-NORM pipeline.

been automatically chosen during the calibration stage, and $p$ specifies the proportion of sentences in which this format is applied. The reformatting step produces a normalized context representation that reduces spurious variability in how evidence is presented to the model. Importantly, the operation is performed at the sentence level, ensuring that semantic content remains intact while surface patterns are harmonized. This guarantees that answer generation relies on content rather than formatting artifacts. Since the same normalization procedure can be applied consistently across different queries, retrieval variations, and domains, the resulting prompts exhibit more uniform structure. Consequently, the target LLM can process long and heterogeneous contexts more effectively, leading to improved robustness and stability in inference-time reasoning.

**To summarize**, C-NORM provides a lightweight, training-free mechanism for adapting context structure to the inductive biases of each model. Instead of requiring parameter updates or additional supervision, it operates purely at the input level by modifying how retrieved content is represented. By aligning the input format with the model's internal dynamics, C-NORM reduces mismatches between surface structure and processing preferences, thereby systematically mitigating brittleness in long-context reasoning. This adjustment not only improves robustness to retrieval noise and ordering impact but also enhances the model's ability to consistently extract relevant information across diverse domains. In effect, C-NORM acts as a compatibility layer between raw retrieval outputs and the target LLM, making downstream reasoning more stable, scalable, and less sensitive to idiosyncratic formatting artifacts.

## 5 EXPERIMENTS

To validate the effectiveness of C-NORM in enhancing the robustness and generalization of LLMs under long-context RAG, we design two complementary evaluation settings: a controlled QA test based on NQ-Open and the real-world task from LongBench-v2 to assess generalizability across diverse input formats and reasoning types. We show that C-NORM consistently improves LLM's long-context reasoning performance over various settings.

### 5.1 CONTROLLED LONG-CONTEXT RAG SETTINGS

In this case, we propose a controlled test using a permuted version of NQ-Open to evaluate both the robustness to order variation and long-context reasoning capacity of LLMs. First, we randomly sample 500 questions from NQ-Open (Liu et al., 2024a). For each question, one gold (relevant) document is identified and mixed with 9 distractors, each containing about 100–300 tokens. We then construct 10 input permutations by placing the gold document at each possible position while shuffling the remaining distractors. The ratio $p$ in C-NORMis fixed at $p = 0.5$ with 8 samples used for selecting the best delimiters.

**Metrics.** We report two complementary metrics. Overall Averaged Accuracy (OAA) measures the accuracy averaged across all gold positions, reflecting robustness to arbitrary permutations. Optimal Positioned Accuracy (OPA) measures the accuracy under the most favorable placement of the gold document, reflecting the model capacity in long-context reasoning regardless of positions. All results are averaged over three random seeds to reduce variance.

**Models.** We adopt several LLMs for evaluation, including LLaMA-2-7B (pretrained context length 4K) (Touvron et al., 2023), LLaMA-2-7B-Chat (4K), Qwen2.5-1.5B (128K) (Yang et al., 2024), and

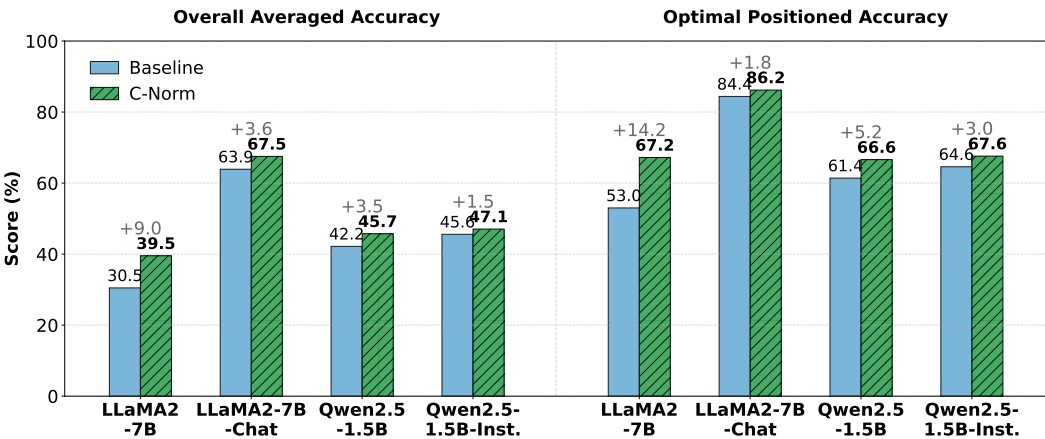

Figure 6: Results on the controlled long-context RAG setting using NQ-Open. We report Overall Averaged Accuracy (OAA) to measure robustness against context order permutations, and Optimal Positioned Accuracy (OPA) to assess capacity under the best placement of the gold document. Baseline denotes the original model, while C-NORM indicates results with contextual normalization.

Qwen2.5-1.5B-Instruct (128K). For base models (e.g., LLaMA-2-7B and Qwen2.5-1.5B), we use unaligned prompts directly following the QA format. For instruction-tuned models (e.g., LLaMA-2-7B-Chat and Qwen2.5-1.5B-Instruct), we adopt aligned prompts that match their chat/instruction interfaces. All generations are performed with temperature fixed at 0 to ensure deterministic outputs and eliminate randomness from sampling.

**Experimental Results.** In the controlled long-context RAG evaluation on NQ-Open, as shown in Figure 6, C-NORM consistently improves both robustness (OAA) and reasoning capacity (OPA) across all evaluated LLMs. The gains are especially pronounced for LLaMA-2-7B, where robustness increases by nearly 30%, showing that format adaptation can compensate for the LLM's limited reasoning ability. It highlight that long-context performance is not only determined by LLM scale or pretraining context window, but also by how the context is presented. Interestingly, the most effective formats are often not the ones most interpretable to humans. For instance, delimiter-heavy or structurally altered representations outperform plain natural text. This underscores the importance of optimizing the input format for alignment with the model's internal dynamics rather than assuming that human-friendly representations are optimal. By automatically selecting a context format that maximizes balanced attention, C-NORM enables models to reason more reliably across arbitrary evidence positions, offering a practical path toward more robust long-context RAG systems.

## 5.2 REAL-WORLD RAG SETTINGS

To evaluate the real-world utility of C-NORM, we adopt LongBench-v2 (Bai et al., 2024), a benchmark targeting long-context reasoning across diverse tasks. It contains 503 multiple-choice questions drawn from six categories, icluding single-document QA, multi-document QA, long in-context learning, dialogue history understanding, codebase comprehension, and structured data understanding. It covers both textual and semi-structured formats. Each question is paired with a long context ranging from 8K to over 2M words, with most falling under 128K, making it ideal for testing long-context generalization. We evaluate under two settings:

- **Base**, where full ground-truth context is given. This design allows us to isolate the effect of C-NORM under partial, noisy, and complete evidence scenarios;

- **RAG**, where top-4 retrieved documents (with retrieval noise) are provided as context, simulating realistic open-domain QA.

We evaluate model performance using overall accuracy, complemented by breakdowns across difficulty levels (Easy and Hard) and context lengths (Short, Medium, and Long). For this setting, we adopt LLaMA-2-7B-Chat and Qwen2.5-1.5B-Instruct, leveraging the official task templates pro-

vided by LongBench to ensure comparability with prior work. To accommodate limited computing resources, we set the maximum prompt length to 4K tokens.

Table 1: Evaluation on LongBench-v2. We report overall accuracy along with breakdowns across difficulty (Easy vs. Hard) and context length (Short, Medium, Long). Baseline denotes the original model, while C-NORM indicates results after applying contextual normalization.

| Model | Setting | Method | Overall | Easy | Hard | Short | Medium | Long |
|---|---|---|---|---|---|---|---|---|
| **LLaMA-2-7B -Chat** | Base | Baseline | 26.4 | 25.0 | 27.3 | 26.7 | **24.7** | 29.6 |
| | | C-NORM | **26.6** | **25.0** | **27.7** | **27.8** | 22.8 | **32.4** |
| | RAG | Baseline | 9.3 | 4.7 | 12.2 | 10.6 | 7.9 | 10.2 |
| | | C-NORM | **10.3** | **5.2** | **13.5** | **11.1** | **8.8** | **12.0** |
| **Qwen2.5-1.5B -Instruct** | Base | Baseline | 23.7 | 24.5 | 23.2 | 29.4 | 21.4 | 18.5 |
| | | C-NORM | **24.7** | **25.0** | **24.4** | **31.1** | **21.9** | **19.4** |
| | RAG | Baseline | 25.6 | **26.6** | 25.1 | **26.1** | 26.0 | 24.1 |
| | | C-NORM | **26.2** | 26.0 | **26.4** | 25.0 | **27.9** | **25.0** |

**Experimental Results.** As presented in Table 1, the results on LongBench-v2 demonstrate that C-NORM consistently improves performance across most metrics, particularly yielding gains in all Hard and Long subsets. This suggests that C-NORM successfully enhances long-context reasoning capabilities without negatively impacting performance in shorter-context scenarios, such as the Easy or Short subsets. The performance gains are more pronounced for LLaMA-2-7B-Chat compared to Qwen2.5-1.5B-Instruct. This can be attributed to the experimental constraint of a 4K-token maximum prompt length, which limits the Qwen's full long-context potential. Nevertheless, the substantial improvements observed for both models under the Long setting underscore the critical role of aligning input formatting with LLM's grounding mechanisms to fully leverage its reasoning capacity in extended contexts.

## 5.3 DISCUSSIONS

While the experiments above demonstrate the effectiveness of C-NORM, several design choices warrant further analysis. In particular, the choice of delimiters and the number of samples used to determine the best delimiter can influence performance and stability. This section discusses these factors, highlighting their practical impact and providing insights for applying C-NORM in different retrieval-augmented generation scenarios.

**Delimiter Choices.** We first examine the effect of delimiter choices in C-NORM. A wider set of candidate delimiters consistently improves performance, as it increases the chance of identifying a format that better aligns with LLM's internal processing. Interestingly, the best-performing delimiter is not always human-interpretable or intuitive. For instance, in our controlled settings, the selected delimiters varies across models: LLaMA-2-7B preferred ".", LLaMA-2-7B-Chat favored ":", Qwen2.5-1.5B chose "-", Qwen2.5-1.5B-Instruct selected "&". Moreover, we observe that the optimal delimiter can also vary across different context settings and lengths, which makes manual selection impractical. These findings underscore two important insights: (1) delimiters that yield high Attention Balance Scores (ABS) can substantially enhance robustness, confirming the effectiveness of C-NORM; and (2) optimal delimiter preferences are both model-specific and context-dependent, highlighting the necessity of automatic selection via ABS rather than relying on human intuition.

**Number of Samples Used for Selecting.** We further examine the sensitivity of C-NORM to the number of samples used when selecting the best delimiter. By varying the sample size from 1 to 10, we observe that the resulting performance and chosen delimiter remain largely stable. This shows that even a very small number of samples is sufficient for reliable delimiter selection, making the procedure computationally efficient. Interestingly, we also find that when the format ratio is varied, the best delimiter may change across settings, indicating that the preferred format is context-dependent rather than determined by token statistics. Combined with the observation in Section 3.2 that attention distributions under C-NORM consistently emphasize central tokens even when the gold document is positioned at the beginning, these results suggest that the gains of C-NORM are

robust and not sensitive to sample size, but rather stem from its ability to adaptively adjust grounding behavior to different context structures.

In summary, our analysis highlights the effectiveness of C-NORM in adapting to diverse models and context settings. The method consistently identifies beneficial delimiters and achieves robust improvements with only a handful of samples. Moreover, the variation of best delimiters across models, context lengths, and format ratios demonstrates the necessity of an automatic, model-guided selection process. These findings underscore that C-NORM provides a lightweight yet powerful mechanism for mitigating positional biases and enhancing grounding, ultimately strengthening long-context reasoning across various LLMs.

## 6 RELATED WORK

**Retrieval Augmented Generation** (RAG) has been widely adopted to improve language models' performance on knowledge-intensive tasks (Borgeaud et al., 2022; Lewis et al., 2020b; Karpukhin et al., 2020). Traditional RAG pipelines usually manage short context windows, typically involving tasks with concise and immediately relevant contexts (Lewis et al., 2020a). While effective for short and well-contained queries, the systems face substantial limitations when scaling to more complex or open-ended tasks (Jeong et al., 2024). Many real-world questions require integrating dispersed evidence from multiple documents or reasoning over lengthy documents such as academic articles, legal cases, or multi-turn dialogues. Standard pipelines that retrieve and concatenate only a few short passages (typically 100–300 tokens each) often suffer from information fragmentation or omission of critical context (Li et al., 2024b; Hsieh et al., 2024). Furthermore, fixed-length context windows in most pretrained LLMs (e.g., 2K–4K tokens) severely limit the amount of retrievable evidence considered simultaneously. These bottlenecks have prompted shifts toward long-context RAG setups, aiming to leverage larger contexts and improved retrieval for open-domain QA (Asai et al., 2023; Lee et al., 2019; Nakano et al., 2021), multi-hop reasoning (Zhong et al., 2023; Ho et al., 2020), and complex document understanding (Dua et al., 2019; Li et al., 2024a).

**Long-Context RAG.** Recent studies (Liu et al., 2024a; Zhang et al., 2024; An et al., 2024; Liu et al., 2024b) have revealed critical limitations in how large language models utilize long-context inputs in RAG. Simply appending more retrieved text does not guarantee improved performance, potentially causing degradation due to positional biases, information dilution, and the "lost-in-the-middle" phenomenon (Liu et al., 2024a). Models often favor content at the beginning or end of the prompt, neglecting relevant information buried in the middle. This results in significant performance variance depending on the order of retrieved documents, even if the overall content remains unchanged (Liu et al., 2024b; Zhang et al., 2024; An et al., 2024). Thus, the effectiveness of long-context RAG is influenced not only by the amount of available information but also by how it is ordered and integrated, motivating a deeper empirical analysis of context-order effects on LLM performance.

## 7 CONCLUSION

In this work, we uncover the overlooked yet critical role of context format in shaping the performance of long-context retrieval-augmented generation. Through systematic analysis, we show that seemingly superficial differences can dramatically shift model accuracy and stability, even when the underlying semantics remain unchanged. To explain this phenomenon, we investigate the mechanisms which underlie the sensitivity of LLMs to how information is structured. Building on these insights, we introduce C-NORM, a lightweight, model-agnostic, and training-free approach that adaptively selects the most effective context format based on the model's own internal dynamics. It provides a simple plug-and-play strategy for standardizing retrieved documents before generation, without requiring architectural changes or additional training overhead. Extensive experiments across both controlled evaluations and the real-world RAG benchmark demonstrate that C-NORM consistently improves the RAG performances. Gains are especially pronounced in challenging long-context scenarios, where retrieval noise and positional biases pose the greatest hurdles.

Ultimately, our findings highlight that reliable grounding in RAG depends not only on what is retrieved, but also on how it is presented to the model. By reframing context presentation as a normalization problem, C-NORM opens a practical new direction for improving the stability and scalability of long-context reasoning in large language models.

## REPRODUCIBILITY STATEMENT

Efforts have been made to ensure the reproducibility of this work. Detailed descriptions of experimental setups, including datasets, preprocessing steps, evaluation protocols, and metrics, are provided in Section 5. The design of controlled settings, such as context permutation and key–value extraction tasks, is specified in Section 2 and further elaborated in Appendix A. Implementation details of C-NORM, including delimiter selection and attention attribution analysis, are described in Section 4. Hyperparameters, such as context lengths, temperature, and ratio $p$, are reported in the respective experimental subsections. To facilitate replication, we will make source code released to public once published.

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

## A   KEY-VALUE EXTRACTION

We adopt a controlled **key–value extraction task** to study the effect of context formatting on retrieval-augmented generation. The task is defined as follows: given a long JSON-like object containing multiple key–value pairs, the model must return the value corresponding to a specified key. Unlike open-domain QA, this setup is free from world knowledge or semantic priors, since both keys and values are synthetic 32-character strings. As a result, performance directly reflects the LLM's ability to utilize and navigate long contexts rather than any memorized information. This task provides a minimal yet effective probe of long-context reasoning. Because all key–value pairs are semantically meaningless, the model cannot rely on prior knowledge; instead, it must depend entirely on the context provided. Success therefore reflects two abilities: (i) robust retrieval under distraction, as the model must locate the gold key among many distractors regardless of position, and (ii) sensitivity to formatting, since any performance difference arises solely from how identifiers are represented (e.g., hyphenated UUIDs versus plain texts). This isolation makes the task particularly well-suited for analyzing how structural cues in the input guide attention and grounding.

We design three variants of the input, differing only in the format of the identifiers:

- **UUID**: Keys and values are expressed as standard universally unique identifiers, represented as 32-character hexadecimal strings with hyphen delimiters.
- **Plain Text**: Identifiers are flattened into continuous 32-character strings without structural delimiters.
- **Modified UUID**: Identifiers are expressed as UUIDs but with hyphens replaced by alternative delimiters (e.g., the "&" symbol).

**Prompt.** The task prompt is shown below. The model is asked to extract the value associated with a given key.

---

**Task Prompt**

**Extract the value corresponding to the specified key in the JSON object below.**

```
# UUID:
550e8400-e29b-41d4-a716-446655440000:
123e4567-e89b-12d3-a456-426614174000

# Plain Text:
550e8400e29b41d4a716446655440000:
123e4567e89b12d3a456-426614174000

# Modified UUID:
550e8400&e29b&41d4&a716&446655440000:
123e4567&e89b&12d3&a456&426614174000
```

**Key:**   xxxxxxx   **Corresponding value:**

---

## B   FREQUENCY-CONTROLLED TOKEN REPLACEMENT

To further analyze whether token exposure during fine-tuning contributes to the observed sensitivity of attention allocation to context format, we design a **Frequency-Controlled Token Replacement** experiment. Specifically, we focus on the Stanford-Alpaca-7B model and construct test cases where context tokens are systematically replaced with tokens of varying frequency in the fine-tuning corpus.

**Settings.** We evaluate the robustness and capacity of LLM on 100 samples from the NQ-Open dataset (Liu et al., 2024a). Each sample is paired with 6 retrieved documents, each containing approximately 100–300 tokens. To simulate long-context reasoning, we permute the position of the gold document across all possible positions. For token replacement, we sort tokens in the Alpaca

fine-tuning data by frequency of occurrence and define replacement groups corresponding to the top-$k\%$ most frequent tokens and bottom-$k\%$ least frequent tokens ($k = 1, 5, 10$). Replacement is enforced by prompting the model to rewrite retrieved passages using only tokens from the allowed set, according to the following instruction:

---

**Replacement Prompt**

**You are given a list of allowed tokens. Your task is to rewrite the text by replacing as many words as possible with the allowed tokens.**

**Rules:**

1. Do **not** add or remove sentences.

2. Do **not** change the order or structure.

3. Only substitute words with allowed tokens when possible.

4. Keep the formatting exactly the same as the original.

**Allowed tokens:**    [token list here]

**Example:**
Original text:   `The cat is sleeping on the mat.`
Rewritten text:   `a cat is sleeping on the mat`

**Now rewrite the following text:**    [original text here]

---

**Results.** Table 2 reports the overall averaged accuracy (OAA) and optimal-position accuracy (OPA) under different replacement groups. The baseline Alpaca model without replacement achieves an OAA of 0.538 and OPA of 0.690. Substituting with frequent tokens (top 10%) slightly reduces performance (OAA = 0.530, OPA = 0.720), while extreme substitution with the most frequent single token further degrades results (OAA = 0.505, OPA = 0.640). Similarly, replacing with least frequent tokens (bottom 10% / 5% / 1%) shows comparable degradation.

Table 2: Performance under frequency-controlled token replacement on NQ-Open with Stanford Alpaca 7B. Top-$k$ and bottom-$k$ indicate substitution using the most and least frequent tokens from the fine-tuning corpus.

| Setting | OAA | OPA |
|---|---|---|
| Stanford Alpaca (no replacement) | 0.538 | 0.690 |
| Top 10% | 0.530 | 0.720 |
| Top 5% | 0.512 | 0.700 |
| Top 1% | 0.505 | 0.640 |
| Bottom 10% | 0.505 | 0.680 |
| Bottom 5% | 0.510 | 0.700 |
| Bottom 1% | 0.502 | 0.670 |

**Discussion.** The results suggest that token frequency alone does not provide a satisfactory explanation for the different attention allocation patterns observed across context formats. Substitutions with both highly frequent and rarely seen tokens lead to similar levels of degradation, and no clear monotonic relationship is observed. This indicates that the grounding mechanism behind context sensitivity is more complex than exposure frequency, and is likely shaped jointly by pretraining dynamics and fine-tuning objectives.

