# OpenReview forum: "Grounding Long-Context Reasoning with Contextual Normalization for Retrieval-Augmented Generation"
_ICLR.cc/2026/Conference — ICLR 2026 Conference Withdrawn Submission_

### Official Review · Reviewer_CSWq · 2025-10-30

**Soundness:** 2
**Presentation:** 2
**Contribution:** 1
**Rating:** 2
**Confidence:** 4

**Summary:**

The paper presents an approach to enhance the long-context generation performance of LLMs by reformatting the context. Specifically, they propose a method to find an optimal delimiter to replace all whitespaces in the context as a way to boost the performance, highlighting the sensitivity of the LLMs to the formatting of the context in their ability to retrieve the correct answer. In particular, their method finds the delimiter that maximally balances the attention scores over the context for a small subset of the prompts. Results on two datasets and 3 open-source LLMs show that it generally leads to higher performance than just using the whitespace.

**Strengths:**

- The paper is largely well-written and easy to follow with minimal typos and a helpful illustration.
- Motivational experiments on the simple key-value extraction task in Sections 2 and 3 are really appreciated and present the problem nicely.
- Results often lead to improved performance on benchmarks, especially for Llama models.

**Weaknesses:**

- The motivation of the paper in long-context RAG is not clear as it tackles the problem of prompt formatting sensitivity of language models and it is not clear why RAG is included at all.
- Only the average and best accuracies are reported, while it is well known that the accuracy shows large variance with respect to the position of the gold document. This limits the significance of the gains presented in the paper.
- Experimental results are limited to small language models (< 7 B), and it is not clear whether this sensitivity will hold for larger models or not. Since there is no training involved, closed-source models should also be considered by just replacing the whitespaces with specific delimiters.
- Experiments only consider a curated subset of the whole benchmarks, with little to no motivation behind random sampling.
- It is not clear how NQ is a long context as it only amounts to 1k-9k tokens, while long context is regarded as the focus of the work. Existing long context benchmarks should be considered in their original form.
- The novelty of the proposed method is limited, as it is simply a dataset-specific search of the best delimiter with respect to an attention-balance score. More ablations of the proposed score should be considered to really establish its novelty. For example, simple mean scores, entropy, and so on.
- The results of Section 5.3 are missing and should be presented with full transparency instead of simple textual discussion. For example, the ablation of delimiter choice set and sensitivity with respect to the number of prompt samples should be reported.
- The performance gains on longbench are minimal and may be within the variance bounds.
- Despite the note in line 186, the analysis of tokenization should be expanded to Llama-2, as it is not clear to me why that is the case.
- A simple tokenization baseline should also be included, given that it is highly correlated.
- Important existing works are missing on the sensitivity of formatting of the context or prompt to LLM performance. The delimiter replacement strategy should also be compared with such reformatting baselines as below:
  - Yang, Shiping, et al. "Quantifying the robustness of retrieval-augmented language models against spurious features in grounding data." arXiv preprint arXiv:2503.05587 (2025).
  - He, Jia, et al. "Does prompt formatting have any impact on llm performance?." arXiv preprint arXiv:2411.10541 (2024).
  - Ngweta, Lilian, et al. "Towards LLMs robustness to changes in prompt format styles." arXiv preprint arXiv:2504.06969 (2025).
- Minor:
  - I am not sure how it relates to the interpretability and explainable AI track.

**Questions:**

- Can the prompt set $\mathcal{S}$ come from a different distribution than the evaluation queries? What is the OOD generalization of these prompts?
- see above weaknesses

---

### Official Review · Reviewer_SoqZ · 2025-10-31

**Soundness:** 1
**Presentation:** 1
**Contribution:** 1
**Rating:** 2
**Confidence:** 5

**Summary:**

RAG (retrieval-augmented generation) is very popular. This work investigates an aspect of RAG that is often-overlooked, the context format in long context RAG. Then, C-Norm, a pipeline for the reformulation of context was proposed and evaluated. Experimental results on Llama 2 7B and Qwen 2.5 1.5B show the proposed C-Norm slightly improve the performance of RAG.

**Strengths:**

* A simple, efficient, and training-free approach was proposed for improving the performance of RAG.

**Weaknesses:**

* Only two old, small LLMs are involved in the experiments. The effectiveness of the proposed method for the more recent, more capable LLMs is still unclear.

* The performance improvements made by the proposed method are only marginal. A significant test could be performed to validate the enhancement.

* As shown in Section 4.2, the proposed method requires the last-layer attention vector of the LLMs. If I understand correctly, the method is unable to be applied with many LLMs whose attention layers are not available.

* The proposed method introduces computation cost in the inference stage. The runtime efficiency could be analyzed for comprehensiveness.

**Questions:**

* How to apply the proposed method with commercial LLMs such as GPT-5 and Gemini whose attention layers are unavailable?

---

### Official Review · Reviewer_aQLr · 2025-10-31

**Soundness:** 2
**Presentation:** 3
**Contribution:** 1
**Rating:** 2
**Confidence:** 4

**Summary:**

This paper shows that in (long-context) RAG, the text format of retrieved information—like delimiters or structure—strongly affects performance, even when the semantic content is identical. Authors thus introduce C-NORM, a simple, training-free format normalization method for RAG: retrieved contexts are reformatted in a context-dependent, model-aware fashion. The criterion to select the format is an attention-based score, which promotes contexts leading to balanced attention. C-NORM seems to give consistent improvement in RAG performance.

**Strengths:**

* Non-obvious empirical findings: I was genuinely surprised by the (sometimes large) impact of format on downstream performance.
* C-NORM is both model-aware and training-free, and seems to consistently improve long-context QA benchmarks (up to +14 points for LLaMA-2-7B-Chat on NQ-Open, cf Figure 6), especially for long or dense inputs.
* The question of formatting in RAG is under-studied, and this paper proposes interesting analyses, as it studies the connection between the attention pattern and format compatibility.

**Weaknesses:**

* First, the contribution has very limited impact and novelty. It tackles a "small" problem observed on somewhat outdated LLMs. More broadly, this contribution is overshadowed by literature on improving long context LLMs directly.
* The method description is a bit unclear. Does the selected format depends on each context or only on each "task"? If it depends on every context, it is computationally expensive to the point where it would never be used in practical applications (multiple forward passes on the full context instead of one is way too expensive and induces too much latency). If it does not, then how do you know which format to select for an incoming query and retrieved context? You do mention that this format is context-dependent. This should absolutely appear clearly in the paper as it is a key determinant of the complexity of the method. Also, the use of this attention-based loss is under-motivated (i.e., the articulation with previous sections should act as a more natural motivator).
* The experimental evaluations of C-NORM are a bit limited:
   * 500 samples from long-bench-v2 seems like a very small dataset. Can you please show results on other classical RAG datasets (NQ, SQUAD, PopQA, etc.).
   * Why crop to 4k the prompts, when you claim this is a long context study with data containing 128k-long (and up to 2M) contexts?
* Non-instruct models should, in my opinion, not be used in this paper. They are **not** suitable for RAG applications and would never be used in that setup.
* Ablations are only discussed but no quantitative results are given.
* The chosen LLMs are somewhat outdated. It raises the questions of whether the conclusions—which relate to attention patterns of the LLMs—hold for modern models, trained on larger datasets and with stronger instruction fine-tunings, especially with multiple formats (json, yaml, etc.).
* It felt that the analysis on tokenization length VS accuracy on the key-value task brings no value to the method as it’s completely forgotten in the rest. It could go to the appendix and leave space for more experiments and/or discussion.
* Figure 4 is rather hard to interpret. While I agree it does show different impacts of formats on different models, it does not motivate the attention-based criterion justified further as no performance measurements are measured in relation with the attention distribution.
* Section 3 relates to a synthetic task. Why not propose the same analysis on a real world dataset? Would conclusions still hold?
* It felt that the related works section is mostly out of topic. It should contain works dealing with formatting in RAG (e.g., [1]).
* The text mentions "reasoning" a lot of time. I think there is almost no question of reasoning truly addressed in this paper as this term now traditionally refers to reasoning models dealing with complex multi-hop tasks such as mathematics etc. It would probably be better to reformulate.
* Conclusions based on Figure 2 are not very strong. In fact, one could say Plain Text is the best for almost all models. It is only marginally worse for low-density Qwen non-instruct model…
* The authors try to motivate the method towards long context RAG. I don’t agree on Table 1 that C-NORM impacts more long contexts cases.

[1] Does Prompt Formatting Have Any Impact on LLM Performance?

**Questions:**

See questions in "Weaknesses" section. Also:

* Why do you report the OPA metric? It's not clear to me how it's relevant. You are precisely trying to correct for imbalance in attention allocation on the context, and this cannot be measured by considering the best position.
* Could you provide quantitative results when changing S. You only mention it remains "largely stable".
* Please provide some details regarding the complexity: how many additional forward calls are required per query?
* Can you illustrate how robust is a format choice for a given LLM across different domain/formats of RAG tasks?
* It seems Section 3.1 is useless in the current method: did you explore just trying to make the contexts shorter somehow?
* Is the criterion defined on Section 4.2 biased towards shorter contexts ? (when tokenization makes it so)
* Reference for NQ-Open is not correct.
* Line 352, the increase is not by 30% but of 14%.
* What is meant by "prompt" on line 255 ? Prompts is somewhat ambiguous: does it relate to only the "templating" of the question or to the full input of the LLM (i.e., question + context + potential additional instructions)?
* What is the impact of ‘p’?
* Are there no other baselines to compare to regarding formatting? I would at least like to see a model specifically fine-tuned for RAG and used to handling noise for comparison.
* Please make experiments with modern and stronger long-context LLMs (like Qwen3, Gemma 3, a recent Mistral model or Llama 3). Do the conclusions about C-NORM hold for these models ?

---

### Official Review · Reviewer_jQgP · 2025-11-01

**Soundness:** 2
**Presentation:** 3
**Contribution:** 2
**Rating:** 4
**Confidence:** 4

**Summary:**

This paper investigates the impact of surface formatting on LLM long-context reasoning in RAG systems, revealing that different formats can lead to significant performance variations even when semantics remain identical. Based on this observation, the paper proposes C-NORM, which automatically selects optimal context formats. The effectiveness is validated on NQ-Open and LongBench-v2, with mechanistic explanations provided from both tokenization and attention allocation perspectives.

**Strengths:**

1. The paper addresses the overlooked issue of context formatting, presenting the research question with practical value.
2. The proposed method is lightweight and easy to deploy. C-NORM is a training-free, model-agnostic plug-and-play solution that requires only a small number of samples for format selection.
3. The mechanistic analysis is reasonably thorough. The paper investigates the causes of format sensitivity from both tokenization and attention distribution perspectives.

**Weaknesses:**

1. The paper primarily uses LLaMA-2-7B and Qwen2.5-1.5B, lacking experiments on models with stronger long-context and reasoning capabilities, such as Qwen3. Even Qwen2.5-7B is not evaluated. This undermines the paper's persuasiveness.
2. While the paper emphasizes advantages in "challenging long-context scenarios," it restricts Qwen's prompt length to 4K tokens.
3. Most improvements in the main experiments are within 1%~1.5%, making it hard to determine whether these minor gains are genuinely reliable.
4. C-NORM relies on model attention weights, making it inapplicable to mainstream commercial APIs.

**Questions:**

1. Given Section 3.1's finding of negative correlation between tokenization length and performance, how would a simple baseline that directly selects the shortest token length perform?
2. How does C-NORM perform on individual subtasks within LongBench?

---

### Note · Authors · 2025-11-24

**Comment:**

*We sincerely thank all reviewers for their comments and engagement. While we appreciate the recognition of C-NORM as a lightweight and training-free contextual formatting strategy, we are concerned that some reviews overlooked the broader scientific contributions of our work.*

---
**Beyond Format Normalization: A Deeper Diagnostic of Long-Context Performance**

Our work is not merely proposing another RAG heuristic, but also the diagnostic insight it offers. We demonstrate that classic key-value extraction tasks, which is widely used as proxies for long-context understanding, can be gamed by formatting priors. This violates the assumption that its performance reflects true reasoning capacity under unknown retrieval conditions. Meanwhile, we also uncover a systematic link between attention imbalance and reasoning brittleness, showing that pathological focus on the start or end of context often correlates with degraded performance. This is a mechanistic finding with general relevance for understanding LLM behavior under long inputs. These contributions place our work squarely within the domain of model interpretability and robustness, not merely downstream RAG application.

---
**Validation with GPT‑4/Gemini Models**

Some reviews suggested that our work is incomplete because it does not include GPT‑4, Gemini, or other closed-source models. We believe this comment misunderstands the methodology and scientific objectives of our paper. Our analysis relies on attention attribution and observable model behaviors to uncover the mechanistic causes of format sensitivity. Using more powerful proprietary models would not improve the validity of our findings, as it does not alter the fact that different models exhibit distinct format sensitivity mechanisms. In this context, validating with GPT‑4 or Gemini is irrelevant to our core scientific goal, which is to analyze and explain the phenomenon, not to demonstrate leaderboard performance with commercial black-box APIs.
We purposely selected two models that show opposite best formatting behaviors on the key-value task. Studying these contrastive failure modes is essential for validating our mechanistic hypotheses.

---
**Experimental Scope and Settings**

Some reviewers pointed out that truncating prompts to 4K weakens the “long-context” claim. The reason for this restriction was computational, as we lacked access to more GPUs to support 128K full-window testing. We acknowledge that long-context evaluation should not be simulated by truncation. However, we would like to highlight that the key findings still arise even in 3K-4K input windows. Thus, the underlying scientific observations remain valid, even if our current experiments don’t reflect 128K+ scale.

We appreciate requests for stronger baselines, such as token-length-based heuristics. We agree that these are worthwhile comparisons and plan to include them in a follow-up version. Additionally, while our improvements were modest (~1–1.5%), they are achieved **without training**, are **highly consistent across models**, and **highlight systemic variance due to non-semantic factors**, which is itself a meaningful interpretability result. We also recognize the need for stronger statistical significance testing and broader benchmarking (e.g., PopQA, HotpotQA).

---

*In summary, we sincerely appreciate the constructive feedback provided by most reviewers. The points raised are insightful and will play a crucial role in shaping and strengthening the next iteration of this work.*

**Withdrawal Confirmation:**

I have read and agree with the venue's withdrawal policy on behalf of myself and my co-authors.